# Deep Neural Networks Improve Radiologists' Performance in Breast Cancer Screening

**Nan Wu**[1], **Jason Phang**[1], **Jungkyu Park**[1], **Yiqiu Shen**[1], **Zhe Huang**[1], **Masha Zorin**[3], **Stanisław Jastrzębski**[4], **Thibault Févry**[1], **Joe Katsnelson**[2], **Eric Kim**[2], **Stacey Wolfson**[2], **Ujas Parikh**[2], **Sushma Gaddam**[2], **Leng Leng Young Lin**[2], **Joshua D. Weinstein**[2], **Krystal Airola**[2], **Eralda Mema**[2], **Stephanie Chung**[2], **Esther Hwang**[2], **Naziya Samreen**[2], **Kara Ho**[2], **Beatriu Reig**[2], **Yiming Gao**[2], **Hildegard Toth**[2], **Kristine Pysarenko**[2], **Alana Lewin**[2], **Jiyon Lee**[2], **Laura Heacock**[2], **S. Gene Kim**[2], **Linda Moy**[2], **Kyunghyun Cho**[1], **Krzysztof J. Geras**[2,1]

[1] *Center for Data Science, New York University*

[2] *Department of Radiology, New York University School of Medicine*

[3] *Department of Computer Science and Technology, University of Cambridge*

[4] *Faculty of Mathematics and Information Technologies, Jagiellonian University*

## 1. Introduction

Breast cancer is the second leading cancer-related cause of death among women in the US. In 2014, over 39 million screening and diagnostic mammography exams were performed in the US. Recent developments in deep learning (LeCun et al., 2015) have opened possibilities for creating a new generation of computer-aided detection tools in mammography.

In this paper[1], we train and evaluate a set of strong neural networks on a mammography dataset of over 200,000 exams (over 1,000,000 images). We use two complimentary types of labels: breast-level labels indicating whether there is a benign or malignant finding in each breast, and pixel-level labels indicating the location of the findings. Our best model, trained on both breast-level and pixel-level labels, achieves an AUC of 0.895 in identifying malignant cases and 0.756 in identifying benign cases on the test set reflecting the screening population. In a reader study, we compared the performance of our best model to that of radiologists and found our model to be as accurate as radiologists in terms of AUC. We also found that a hybrid model, taking the average of the probabilities of malignancy predicted by a radiologist and by our neural network, yields more accurate predictions than either separately. Finally, we have published the code and weights of our best models online.

## 2. Deep CNNs for cancer classification

**Data**   Our dataset (Wu et al., 2019) includes 229,426 digital screening mammography exams. Each exam was assigned labels indicating whether each breast was found to have biopsy-proven malignant or benign findings. We have 5,832 exams with at least one biopsy performed within 120 days of the screening mammogram. Among these, 985 breasts had malignant findings, 5,556 breasts had benign findings and 234 breasts had both malignant and benign findings. For all exams matched with biopsies, we asked a group of radiologists to retrospectively indicate the location of the biopsied lesions at a pixel level.

---

1. This is a shorter version of the paper of the same title available at https://arxiv.org/pdf/1903.08297.pdf.

**Breast-level cancer classification model**  We trained a CNN to produce four predictions corresponding to the four labels for each exam given the four screening mammography views. Figure 1 provides an overview of the model's inputs, outputs and architecture. The overall network consists of two modules: (i) four view-specific columns, each based on the ResNet architecture[2] (He et al., 2016) that outputs a fixed-size representation, and (ii) two fully connected layers to map these representations to predicted probabilities. Weights are shared between the L-CC and R-CC columns, as well as the L-MLO and R-MLO columns. We average the probabilities predicted by the CC and MLO branches of the model to obtain the final predictions.

**Patch-level classification model**  We trained an auxiliary model to classify $256 \times 256$-pixel patches of mammograms, predicting the presence or absence of malignant and benign findings in a given patch. The labels for these patches are produced based on overlap with the pixel-level segmentations. We then apply this auxiliary model to the full resolution mammograms in a sliding window fashion to create two 'heatmaps' for each image (Figure 2), containing the estimated probability of malignant and benign findings within a corresponding patch. These heatmaps are used as additional input channels to the breast-level model to provide supplementary fine-grained information. This approach allows us to use a very deep auxiliary network—a DenseNet121 (Huang et al., 2017)—on the patches, initialized from pretraining on large off-domain data sets such as ImageNet (Deng et al., 2009).

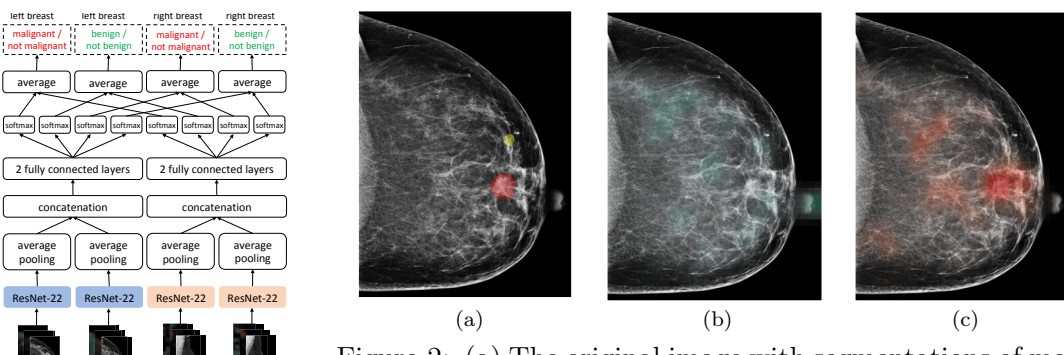

Figure 1: Breast-level model.

Figure 2: (a) The original image with segmentations of malignant and benign findings. (b) A heatmap for benign findings. (c) A heatmap for malignant findings.

**Experiments**  In all experiments, we used the training set for optimizing parameters of our model and the validation set for tuning the hyperparameters of the model and the training procedure. To further improve our results, we applied model ensembling (Dietterich, 2000), wherein we trained five copies of each model with different random initializations of the weights in the fully connected layers. The ResNet weights are initialized with the weights of the model pretrained on BI-RADS classification (Geras et al., 2017).

**Results**  Results are reported on the test set, which approximates the population undergoing routine screening. The model ensemble using only the original images and no heatmaps achieved an AUC of 0.840 for malignant/not malignant classification and an AUC of 0.743 for

---

2. *ResNet-22* in Figure 1 refers to a 22-layer ResNet based on the ResNet architecture with additional modifications such as a larger kernel in the first convolutional layer and fewer filters in each layer.

benign/not benign classification. The model ensemble using both the original images and the heatmaps achieved an AUC of 0.895 for malignant/not malignant and 0.756 for benign/not benign classification, outperforming the image-only model on both tasks. The difference in performance of our models between these two tasks can be largely explained by the fact that a larger fraction of benign findings than malignant findings are mammographically-occult (cf. Table 2 in (Wu et al., 2019)).

## 3. Reader study

To compare the performance of our image-and-heatmaps ensemble (*the model*) to human radiologists on malignancy classification, we performed a reader study with 14 radiologists with varying levels of experience, each reading 720 exams from the test set and providing a probability estimate of malignancy for each breast. Among the 1,440 breasts from 720 exams, 62 breasts are labeled as malignant and 356 breasts are labeled as benign. On this subpopulation, our model achieved an AUC of 0.876, while AUCs achieved by individual readers varied from 0.705 to 0.860 (cf. left panel in Figure 3). We also evaluated the accuracy of a human-machine hybrid, whose predictions are the averaged predictions of a radiologist and of the model. Hybrids between each reader and the model achieved an average AUC of 0.891 (std: 0.0109) (cf. middle panel in Figure 3). These results suggest our model can be used as a tool to assist radiologists in reading breast cancer screening exams and that it may capture different aspects of the task compared to experienced breast radiologists.

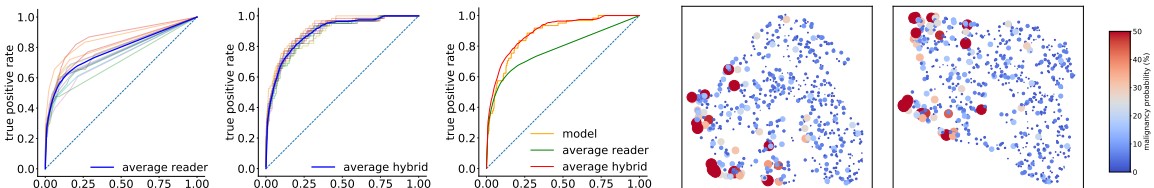

Figure 3: ROC curves of exams in reader study.     Figure 4: Visualization of activations.

In Figure 4, we visualize two sets of activations for each exam by embedding them into a two-dimensional space using UMAP (McInnes et al., 2018): concatenated activations from the last layer of each of the four image-specific columns (left), and concatenated activations from the first fully connected layer (right). The color and size of each point reflects the same information: the warmer and larger a point is, the higher the readers' mean prediction of malignancy is. We observe that exams classified as more likely to be malignant according to the readers are close to each other for both sets of activations. The fact that previously unseen exams with malignancies were found by the network to be close in this low-dimensional space further corroborates that our model exhibits strong generalization capabilities.

## 4. Discussion

By leveraging a large dataset with breast-level and pixel-level labels, we built a neural network which can accurately classify breast cancer screening exams. We showed that a hybrid model including both a neural network and expert radiologists outperformed either individually. This suggests that the use of such a model could improve radiologist sensitivity for breast cancer detection.

## Acknowledgments

The authors would like to thank Catriona C. Geras for correcting earlier versions of this manuscript, Michael Cantor for providing us pathology reports, Marc Parente and Eli Bogom-Shanon for help with importing the image data and Mario Videna for supporting our computing environment. We also gratefully acknowledge the support of Nvidia Corporation with the donation of some of the GPUs used in this research. This work was supported in part by grants from the National Institutes of Health (R21CA225175 and P41EB017183).

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
