# OpenReview forum: "Deep Neural Networks Improve Radiologists' Performance in Breast Cancer Screening"
_MIDL.io/2019/Conference/Abstract — MIDL Abstract 2019_

### Official Review · AnonReviewer1 · 2019-04-30
**Deep Neural networks trained on pixel and breast level labels are shown to help radiologists in their decisions**

**Rating:** 4
**Confidence:** 2

**Review:**


Summary:

In this work, a deep learning method based that is trained on breast- and pixel-level labels is presented used to classify breast cancer screenings. Experiments show that the neural network model can be used to assist radiologists to make improved classifications.

Comments:

+ Good utilisation of patch level classification model to obtain the heatmap representations
+ Results showing improvement in the hybrid model (network + radiologists) is very interesting and can be of great interest to the audience at MIDL.
+ Experiments on the large dataset are thorough and the paper is clearly written
+ Nice that the code and networks are made available

- A concise abstract even in this length of the paper can be useful to orient the readers

---

### Official Review · AnonReviewer2 · 2019-04-30
**Multipathway deep learning approach with interesting experimental results**

**Rating:** 3
**Confidence:** 2

**Review:**

Breast cancer screening is addressed by a multipathway model that inputs two views per each breast into a ResNet-22 architecture. The architecture shares weights for the left-right views. The prediction probability for malignant or benign structures are obtained by averaging the output probabilities. In the proposed hybrid model, the authors included in the average also the prediction score of a radiologist (or a pool of radiologists, not clear from the manuscript). Experiments showed that the hybrid model had best performance against the reference biopsy-proven malignant or benign breast mammography scan classifications.

This manuscript is a short version of a paper published on arXiv.

Pros:
- Experiment design seems solid and results are concisely reported.
- The finding that averaging the prediction scores output by the model and a trained radiologist is somewhat surprising, given that the range of AUCs of the radiologists was rather wide. Some insightful discussion on this finding would further strengthen the contribution.

Cons:
- In Section 3, the statement that a radiologist provided “a probability estimate of malignancy for each breast.” How this probability was assigned and interpreted is unclear. In subsequent sentences the authors state that certain number of breast was labeled malignant, etc. Were the labeles defined based on the assigned probabilites or were the radiologists indeed labeling the images?
- The proposed hybrid model included the average prediction scores of the CNN model and the radiologst. It is not clear from the manuscript whether this score was taken consistently from one particular radiologist or a pool of radiologists?

---

### Decision · Program_Chairs · 2019-05-06
**Acceptance Decision**

Accept